# Efficient Separation and Recovery of Petroleum Hydrocarbon from Oily Sludge by a Combination of Adsorption and Demulsification

**DOI:** 10.3390/ijms23147504

**Published:** 2022-07-06

**Authors:** Mingzhu Yao, Yun Ma, Lu Liu, Chengrong Qin, Haibo Huang, Zhiwei Zhang, Chen Liang, Shuangquan Yao

**Affiliations:** Guangxi Key Laboratory of Clean Pulp & Papermaking and Pollution Control, School of Light Industrial and Food Engineering, Guangxi University, Nanning 530004, China; yaomingzhu@st.gxu.edu.cn (M.Y.); 1916401003@st.gxu.edu.cn (Y.M.); gxdxll123456@163.com (L.L.); hebo1104@outlook.com (H.H.); zzw15189837288@163.com (Z.Z.); liangchen@gxu.edu.cn (C.L.)

**Keywords:** oily sludge, sodium lignosulfonate, adsorption demulsification, separation, recovery

## Abstract

The treatment of oily sludge (OS) can not only effectively solve environmental pollution but also contribute to the efficient use of energy. In this study, the separation effect of OS was analyzed through sodium lignosulfonate (SL)-assisted sodium persulfate (S/D) treatment. The effects of SL concentration, pH, temperature, solid–liquid ratio, revolving speed, and time on SL adsorption solubilization were analyzed. The effects of sodium persulfate dosage, demulsification temperature, and demulsification time on sodium persulfate oxidative demulsification were analyzed. The oil removal efficiency was as high as 91.28%. The results showed that the sediment was uniformly and finely distributed in the S/D-treated OS. The contact angle of the sediment surface was 40°, and the initial apparent viscosity of the OS was 56 Pa·s. First, the saturated hydrocarbons and aromatic hydrocarbons on the sediment surface were adsorbed by the monolayer adsorption on SL. Stubborn, cohesive oil agglomerates were dissociated. Sulfate radical anion (SO_4_^−^·) with a high oxidation potential, was formed from sodium persulfate. The oxidation reaction occurred between SO_4_^−^· and polycyclic aromatic hydrocarbons. A good three-phase separation effect was attained. The oil recovery reached 89.65%. This provides theoretical support for the efficient clean separation of oily sludge.

## 1. Introduction

Oily sludge (OS) is a type of solid waste generated by oil extraction, refining, storage, and transportation [1]. It is classified into landing oil sludge, oil-based drilling chips, refining/chemical sludge, and bin/tank bottom oil sludge [2]. Oily sludge is a stable water-in-oil (W/O) emulsion system composed of water, solids, petroleum hydrocarbons (PHCs), and metal ions [3]. The output of OS has been increasing with the increasing global demand for oil resources. OS contains high concentrations of PHCs, heavy metals, sulfides, benzene, anthracene, phenanthrene, phenol, and other toxic and hazardous substances. PHCs include saturated hydrocarbons, polycyclic aromatic hydrocarbons, asphaltenes, and resins. These compounds may infiltrate the soil and groundwater, forming a ground cover that hinders the growth of vegetation, which is toxic and persistent, posing a serious threat to the environment and human health [4]. At present, OS is classified as hazardous waste in many countries, and the treatment of OS has attracted increasing attention [5,6]. The treatment methods mainly include solvent extraction [7,8], pyrolysis [9], electrochemical technology [10], chemical thermal washing [11], and quenched and tempered centrifugal separation [12]. As a mainstream and mature technology, thermal washing is widely used for the separation of OS. The separation of OS is realized by the contact replacement and adsorption of chemicals [13]. Therefore, the research, development, and application of thermal washing are of great significance.

Thermal washing is mainly based on the solubilization and crimping effect of surfactants to separate PHCs from OS [14]. In recent years, many studies have been conducted on chemical thermal washing [15], the selection of surfactants [16], and the stability of emulsions [17]. The surface tension of the oil–water interface has been reduced by adding surfactants [18], and the migration of PHCs has been facilitated [19]. This contributed to the solubilization of micelles and promoted the removal of PHCs [20]. However, traditional chemical surfactants are toxic, difficult to degrade, and cause secondary pollution to the environment [21,22]. Therefore, it is important to develop new green and efficient surfactants. Sodium lignosulfonate (SL) is an anionic surfactant that is a by-product of sulfite cooking [23]. It has strong hydrophilicity and a hydrophobic structure. The separation of OS using SL was investigated in our previous study. The oil removal efficiency was higher by 12.86% compared to using traditional sodium dodecyl sulfonate [24,25]. However, SL is a randomly branched polymer and differs from ordinary surfactants in that it cannot form a regular phase interface state. No particularly clear critical micelle concentration has been found [26]. Experimental results showed low oil–water separation efficiency, low oil absorption capacity, and a poor three-phase separation effect in the SL treatment. In fact, the demulsifier could quickly reach the oil–water interface and replace the original substance at the interface. Thus, the interface strength was reduced and droplet coalescence was promoted. The W/O emulsion could be effectively separated in OS [27]. Sodium persulfate is a typical demulsifier. Bouzid et al. [28] analyzed the kinetics and selectivity of the oxidation of polycyclic aromatic hydrocarbons (PAHs) by persulfate. The results showed that larger PAHs were not difficult to demulsify in micelles. Lominchar et al. [29] analyzed the degradation characteristics of aged diesel oil in soil using persulfate. The removal efficiency of total PHCs in the soil was 98%. Therefore, the effect of the adsorption demulsification separation of OS through the organic combination of SL and sodium persulfate is of great significance for further research. In fact, the main advantage of oily sludge separation is to solve soil pollution. The treated soil can be directly used in industrial and agricultural production. In addition, the separated oil can be recycled. As a by-product of biomass refining, sodium lignosulfonate is used to heat wash oily sludge. It does not enhance the added value of lignocellulosic biomass, but has high environmental value and economic feasibility.

In this study, OS was efficiently separated by SL-assisted sodium persulfate (S/D) treatment. The synergistic effects of SL and sodium persulfate were also analyzed. The effects of pH, SL concentration, temperature, solid–liquid ratio, revolving speed, and SL treatment time on the oil removal efficiency were analyzed. The W/O emulsion containing oil components was demulsified using sodium persulfate. The effects of sodium persulfate dosage, demulsification temperature, and demulsification time on the oil removal efficiency were analyzed. The efficiency of OS separation by the traditional SL treatment and sodium persulfate (DL) treatment were compared. The apparent properties were analyzed through scanning electron microscopy (SEM), contact angle (CA), Fourier transform infrared spectroscopy (FTIR), and viscosity measurements. The effects of different treatments on the removal of PHCs from OS were analyzed through infrared thermogravimetric Fourier transform infrared spectroscopy (TG-FTIR). These results provide theoretical support for the efficient and clean separation of OS.

## 2. Results and Discussion

### 2.1. Effect of Adsorption on Oil Removal Efficiency

The PHCs were separated by SL treatment. However, the three-phase separation of oil, water, and soil was difficult to achieve. This was attributed to the fact that the OS contained a certain amount of water, and a complex W/O emulsion formed. The three-phase separation was blocked [30]. The demulsification effect of sodium persulfate was utilized in the sludge treatment process [31]. In this study, the efficient separation of OS using S/D treatment was analyzed.

The adsorption of SL on OS was analyzed under fixed sodium persulfate treatment conditions. The sodium persulfate dosage was 3.0%. First, the effect of pH (8, 9, 10, 11, 12, 13, and 14) on the oil removal efficiency was analyzed. Other conditions were as follows: SL concentration of 2.0 g·L^−1^, adsorption temperature of 30 °C, the solid-to-liquid ratio of 1:30, revolving speed of 200 rpm, and SL treatment time of 3.0 h. The results are presented in Figure 1a. The oil removal efficiency gradually increased from 56.59% to 61.04% with the increase in pH from 8 to 11. This was attributed to the increase in the ionization of hydrophilic groups (sulfonic, carboxyl, and phenolic hydroxyl groups) in SL molecules with the increase in pH. The SL charge density increased, and its adsorption of PHCs improved [32]. The PHCs were removed by solubilization and crimping. The oil removal efficiency increased significantly as the pH increased from 11 to 13 (75.10% at pH 13). This was attributed to the increase in the ionization degree of carboxyl and phenolic hydroxyl groups with an increase in pH. The activity and solubilization of SL also improved. However, the increase in the oil removal efficiency did not change significantly with the continuous increase in pH. The surface tension remained unchanged, resulting in a stable oil removal efficiency of OS [33]. Therefore, the optimum pH was determined to be 13.

Subsequently, the effect of the SL concentration (0.5, 1.0, 1.5, 2.0, 2.5, and 3.0 g·L^−1^) on the oil removal efficiency was analyzed under the optimum pH. Other conditions were the same. The results are shown in Figure 1b. The oil removal efficiency was 66.93% at 0.5 g·L^−1^ SL and increased with an increase in SL concentration. The maximum oil removal efficiency (80.31%) was obtained at 2.5 g·L^−1^ SL. SL is an amphiphilic anionic surfactant; a low concentration of SL molecules aggregated on the oil−solid interfacial film, thus changing the wettability of the interface [14]. In addition, the SL hydrophilic groups repelled the PHCs in the aqueous phase, and the removal of PHCs from the silt surface was promoted. SL micelles formed with an increase in concentration, and PHCs were adsorbed onto the core of the hydrophobic micelles, as shown in Figure 1g. A microemulsion was then formed. Eventually, it was removed from the sludge surface. The oil removal efficiency increased with the solubilization of micelles. The separation of PHCs was inhibited when the SL concentration exceeded the critical micelle concentration. The oil removal efficiency decreased by 5.0% at 3.0 g·L^−1^ SL. This was because of the formation of numerous microemulsions [34]. The fluidity of PHCs decreased with an increase in the microemulsion concentration, and it was difficult for the PHCs on the sludge surface to be adsorbed [35]. Therefore, the optimum SL concentration was determined to be 2.5 g·L^−1^.

The SL charge largely depends on the temperature [36]. Tang et al. [37] found that temperature has a significant effect on the rheological properties and SL ionization degree. Therefore, the effect of the adsorption temperature (20, 30, 40, 50, and 60 °C) on the oil removal efficiency was analyzed. The results are shown in Figure 1c. The oil removal efficiency increased from 74.35% to 84.56% as the temperature increased from 20 to 40 °C. This illustrates that temperature had an extreme influence on the solubility of PHCs. The thermal motion of the molecules and, hence, the apparent water solubilities of hydrophobic pollutants (such as phenanthrene, pyrene, etc.) increased with an increase in the temperature of the water system. The solubility of ionic surfactants also increased with an increase in temperature [38]. Numerous micelles in the system were formed at the Krafft point. However, the oil removal efficiency decreased at temperatures higher than 40 °C (72.93% at 60 °C). This was attributed to the fact that the temperature exceeded the Krafft point of SL, and the micelles dissolved and decreased the demulsification effect [39]. Therefore, the optimum reaction temperature was 40 °C.

OS has poor fluidity and high viscosity. Duan et al. [40] found that the volume of the liquid has a significant effect on the fluidity of OS. Therefore, the effect of the liquid-to-liquid ratio (1:15, 1:20, 1:25, 1:30, and 1:35) of the reaction system on oil removal efficiency was analyzed. The results are presented in Figure 1d. The oil removal efficiency increased from 80.24% to 86.37% with an increase in the solid–liquid ratio. This was attributed to an increase in the adsorption efficiency with the SL molecular diffusion coefficient. SL diffused rapidly between the surface and interface of oily sludge. A large amount of PHCs were removed because of the increase in solubilization [41]. However, it decreased from 86.37% to 79.83% at a solid–liquid ratio of 1:35. This was attributed to an O/W emulsion formed in the system with an increase in the liquid volume, and the emulsifying properties of the liquid increased [42]. The adsorption demulsification effect was weakened owing to the decrease in the contact area of SO_4_^−^· with the oil phase, thus decreasing the removal of PHCs. Therefore, the optimum solid–liquid ratio was 1:25.

The revolving speed is an important factor affecting the degree of contact between the surfactant and OS; thus, its effect on oil removal was analyzed. The values studied were 50, 100, 150, 200, and 250 rpm, and the results are shown in Figure 1e. The oil removal efficiency was 80.01% at 50 rpm and increased to 87.04% at 150 rpm, the maximum oil removal efficiency obtained. This was attributed to the increase in collisions between solid particles and surfactants in the sludge with an increase in revolving speed [43]. The effective contact between the PHCs and SL improved, promoting the removal of PHCs. However, it decreased to 84.59% at 250 rpm. In fact, full emulsification of OS was realized at a high speed. A stable O/W emulsion was formed after the removal of PHCs [15]. After emulsification, the OS particles were more delicate, and their effective sedimentation was difficult to achieve. Therefore, the optimum revolving speed is 150 rpm.

It took a certain amount of time to form micelles after the contact between the SL and OS in the system [44]. The removal difficulty varied with different PHCs. Therefore, the effect of SL treatment time (1.5, 2.0, 2.5, 3.0, and 3.5 h) on the oil removal efficiency was analyzed, and the results are shown in Figure 1g. It increased from 70.96% to 88.91% as the treatment time increased from 1.5 h to 3.0 h. However, it decreased when the SL treatment time was longer than 3.0 h. This was attributed to the strong adsorption forces between the PHCs and sludge surface. Some of the removed heavy oil adsorbed on the soil surface again under the action of a long-term shear force. This is called the return phenomenon [45]. The effective reaction time must be controlled for the effective removal of PHCs from OS. The optimum SL treatment time was 3.0 h.

The optimum conditions of SL treatment in adsorption demulsification were 2.5 g·L^−1^ SL concentration, pH 13, temperature of 40 °C, solid–liquid ratio of 1:25, revolving speed of 150 rpm, and treatment time of 3.0 h. The oil removal efficiency under these conditions was 88.91%.

The schematic diagram of treating oily sludge with SL was shown in Figure 1f. The untreated oily sludge was brownish black, the surface of the sediment (brownish black) deposited at the bottom of the beaker was covered with a large amount of oil agglomeration (yellow), as shown in the enlarged circle on the left side of the beaker. In addition, a small amount of water (blue) was coated inside the oil, forming W/O particles, as shown in the enlarged square on the right side of the beaker. After SL treatment, the oily sludge became lighter in color (the beaker below was brownish gray) due to the dissolution of a large amount of oil. The oil on the surface of the sediment deposited at the bottom of the beaker was largely removed. In addition, a large number of micellar systems coated with SL and many fine W/O particles were formed in the upper water layer.

### 2.2. Effect of Demulsification on Oil Recovery

The results showed that PHCs were removed from the sludge after SL treatment. Eventually, a typical W/O formed. This means that a three-phase separation could not be achieved. Thus, the demulsification effect of sodium persulfate after SL treatment was analyzed. The effects of sodium persulfate dosage, demulsification temperature, and demulsification time on oil removal efficiency were analyzed.

Mora et al. [46] showed that sodium persulfate can effectively degrade phenanthrene in soil and realized the separation of OS. First, the effect of sodium persulfate dosage (1.0, 3.0, 5.0, 7.0, and 9.0%) on the oil removal efficiency was analyzed. The demulsification temperature was 40 °C, and the demulsification time was 30 min. The results are presented in Figure 2a. It increased from 79.28% to 88.97% as the sodium persulfate dosage increased from 1.0% to 5.0%. In fact, SO_4_^−^·, a non-selective oxidant with a high redox potential (E_0_ = 2.6 v) [29], formed from sodium persulfate activation. PHCs were oxidized and degraded by SO_4_^−^· Organic compounds such as alkanes, alcohols, organic acids, ethers, and esters in oily sludge were decomposed into small molecules. Thus, the removal of PHCs was promoted. However, it decreased from 88.97% to 80.17% at a sodium persulfate dosage of 9.0%. This was because a large amount of SO_4_^−^· and OH· free radicals quickly formed in the excess sodium persulfate solution. It was quenched to failure as the high local concentration of SO_4_^−^· promoted mutual free radicals [47]. The eluted heavy oil was re-adsorbed onto the solid surface. Therefore, the optimum sodium persulfate dosage was 5.0%.

OS emulsion is a complex and stable liquid–liquid colloid suspension system. Its demulsification process is a complex process [48]. The densification temperature is one of the main factors affecting demulsification. Therefore, the effect of the demulsification temperature (40, 50, 60, 70, and 80 °C) on the oil removal efficiency was analyzed at the optimum sodium persulfate dosage and the demulsification time of 30 min. The results are presented in Figure 2b. It increased from 80.29% to 91.35% as the temperature increased from 40 °C to 80 °C. It remained unchanged at temperatures above 80 °C. This was attributed to the inhibition of the formation of SO_4_^−^· by the low activity of S_2_O_8_^2−^ at low temperature. The peroxide bond of sodium persulfate was broken through the adsorption of heat energy, and the formation of SO_4_^−^· was promoted [31]. It could react, not only with the compounds with unsaturated bonds (such as olefins and alkynes), but also with organic compounds with electron donor substituents (such as amines, amines, and hydroxyl groups) [49]. The emulsion was then broken, and the effect of the three-phase separation improved. However, the radical quenching reaction accelerated at higher temperatures. Therefore, the optimum demulsification temperature was 70 °C.

Yuan et al. [50] found that the demulsification time of persulfate has a significant effect on the removal of phenanthrene from sludge. Therefore, the effect of the demulsification time (10, 30, 60, 90, and 120 min) on the oil removal efficiency was analyzed, and the results are shown in Figure 2c. It increased from 73.89% to 91.28% as the demulsification time increased from 10 to 60 min. The activation of S_2_O_8_^2−^ and the reaction of SO_4_^−^· with PHCs requires a certain amount of time; therefore, it increases with time [28]. However, it decreased gradually when the demulsification time exceeded 60 min (80.98% at 120 min). This was attributed to the increase in the re-adsorption of PHCs on the sludge surface under the action of long−term shear force [27]. Therefore, the optimal demulsification time was 60 min.

On the basis of the adsorption of SL, the oily sludge was obviously separated by demulsification of sodium persulfate (Figure 2d). A large number of W/O particles in the upper water layer of the beaker are broken by SO_4_^−^·, as shown in the square on the left side of the beaker. The coalescence of oil droplets was promoted, and the oil–water separation was achieved.

In summary, the optimal conditions for S/D treatment were an SL concentration of 2.5 g·L^−1^, pH 13, adsorption temperature of 40 °C, solid–liquid ratio of 1:25, rotational speed of 150 rpm, adsorption time of 3 h, sodium persulfate dosage of 5.0%, demulsification temperature of 70 °C, and demulsification time of 60 min. The oil removal efficiency of the OS was as high as 91.28% under these conditions. It increased by 15.69% and 29.01% compared with the traditional SL and DL treatments, respectively. This means that S/D treatment is a new method for the efficient separation of OS.

### 2.3. Apparent Morphology and Physicochemical Properties of Oily Sludge

The surface morphology of the OS after different treatments was analyzed. Figure 3a shows that a large number of aggregates existed in the original OS. The surface of the soil was bright because of the adsorption of a large amount of PHCs. There was no significant difference between the blank control experiment (OS’) and OS. The oil content was 36.30% and 31.20%, respectively. This meant that the PHCs on the surface of the OS could not be effectively removed by simple hydraulic shearing. Fine sand particles on the surface of the OS were found after the SL treatment. This indicates that the stable interface between the sludge and PHCs was destroyed. The irregular distribution of soil particles was exposed. Part of the highlighted block structure was caused by the unremoved PHCs [51]. The oil content was 8.86%. The sludge with DL treatment dispersed more soil particles. This indicated that the demulsifier of sodium persulfate destroyed the stable W/O emulsion system of the OS. The PHCs were removed via oxidation, and the oil content was 13.70%. Loose and uniformly distributed silt particles were found in the sludge after S/D treatment. This indicates that the heavy oil surface film was destroyed by SL curling solubilization. A large amount of PHCs were removed and dissolved in the SL solution. The W/O structure was broken for the demulsification effect of sodium persulfate and the separation of oil and water was promoted [52]. The oil content was as low as 6.93%. Thus, a good separation of the OS was obtained by S/D treatment. The results were confirmed using CA analysis.

The contact angles of the OS after different treatments were analyzed. The results are shown in Figure 3b. The wettability of a treated soil surface is an important index of the oil removal efficiency of OS [53]. The surface contact angles of OS and OS’ were 129° and 125°, respectively. This showed a strong hydrophobicity of the original OS and the OS without a cleaning agent. However, the CA of the OS decreased after thermal washing. The CAs of SL and DL were similar (85° and 88°, respectively). This means that the separation effects of the traditional SL treatment and DL treatment were very similar. The contact angle of the S/D-treated sludge significantly decreased (40°). The sludge exhibited a strongly hydrophilic surface. Efficient separation of the OS was obtained.

OS is a type of multiphase emulsion. It contains a variety of strong polar natural emulsifiers, including asphaltene and resin. The stability and viscosity of the OS are enhanced owing to the aggregation of asphaltene molecules and their interaction with resin [54]. Therefore, viscosity is another important index of the oil removal effect. The sludge was a pseudoplastic fluid, and the apparent viscosity of the fluid decreased with an increase in the shear yield [55]. The viscosities of the different OS are shown in Figure 3c. The initial viscosity of the OS was the highest (1500 Pa·s). OS’ had a similar initial viscosity of 1244 Pa·s. This indicates that physical shear had no obvious effect on the viscosity of OS without a cleaning agent. The initial viscosity of the SL-treated sludge was lower (680 Pa·s). This showed that the molecular interaction between heavy components (such as asphaltenes and colloids) was weakened. The heavy oil was diluted [56]. Therefore, the viscosity of the OS decreased. In addition, the oil content in the DL-treated sludge was higher than that in the SL-treated one. However, its initial apparent viscosity was only 72 Pa·s. This was attributed to the effective penetration and disintegration of the bonded asphaltene film network because of the addition of a demulsifier. The interfacial tension and viscoelastic shear modulus of the interfacial film decreased, and the agglomeration of water droplets was promoted [57]. Saturated hydrocarbons and aromatic light petroleum hydrocarbons were removed by SL treatment. It had a weak crimping effect on the asphalts and colloids with high viscosity. The network of the asphalt interfacial film was effectively destroyed by sodium persulfate. This illustrated that the viscosity of the OS emulsion was mainly related to the heavy components, such as resin and asphaltene [58]. It played an important role in the stability of the O/W emulsion. The S/D-treated sludge had the lowest oil content (6.93%), and its initial viscosity was 56 Pa·s. This illustrated that efficient oil removal and effective viscosity reduction were obtained under the combined effect of adsorption and demulsification.

The changes in the main functional groups in OS after treatment were analyzed by FTIR, and the results are presented in Figure 3d. In the OS and OS’ samples, the stronger absorption peak at 3437 cm^−1^ is attributed to the stretching vibration of –OH and –NH bonds in PHCs. The peak at 2960 cm^−1^ is the stretching vibration of the –CH_3_ of alkanes. There is a stretching vibration of –CH_2_ at 2917 cm^−1^ and 2847 cm^−1^. The stretching vibrations of the benzene ring backbone C=C in aromatic or mononuclear aromatic hydrocarbons are observed at 1620 cm^−1^ and 1460 cm^−1^, respectively. The C–O–C stretching vibrations of alicyclic ethers are observed at 1130 cm^−1^ and 1000 cm^−1^. The weak absorption peaks at 780 cm^−1^ and 860 cm^−1^ are attributed to benzene derivatives. Therefore, the original OS mainly consists of alkanes, olefins, and aromatic hydrocarbon compounds [59]. The stretching vibration peaks of alkynes disappear at 3300–3040 cm^−1^ in the sludge after different treatments. This indicated that the olefins in the OS were effectively removed by the adsorption of SL or the emulsion breaking of sodium persulfate. However, the stretching vibration peaks of alkane –CH_3_ and –CH_2_ are reduced at 2960 cm^−1^, 2917 cm^−1^, and 2847 cm^−1^ in the S/D-treated sludge. This showed that the capability to remove alkanes was increased in the S/D treatment. In addition, there are two new absorption peaks at 1460 cm^−1^ and 1355 cm^−1^. They are assigned to the stretching vibrations of isopropyl and butyl C–H, respectively. This indicated that PAHs in the OS were broken down into small molecules by the S/D treatment [60]. Therefore, it was not only effective for saturated hydrocarbons and aromatic ring oils and fats, but also effectively decomposed asphaltenes and resins.

### 2.4. Change of Composition of PHCs from the Oily Sludge

The gas products were analyzed using TG-FTIR during the thermal decomposition of OS. The PHC components in the OS were analyzed. Figure 4 shows the thermogravimetric analysis, infrared spectra, and 3D-thermogravimetric infrared images of the volatile pyrolysis products at different temperatures. PHCs were decomposed into gaseous products, such as H_2_O and CO_2_, during pyrolysis. The weight loss yield in the OS pyrolysis was 54.17%, as shown in Figure 4a. The pyrolysis weight loss yield of OS’ was 51.15%. This was attributed to the poor effect of hydraulic shearing alone on the separation of OS. The efficiencies of the pyrolysis of the OS treated by SL and DL are 43.77 and 38.69%, respectively. This was not consistent with the oil removal efficiency of OS after different treatments. DL treatment had a strong demulsification effect, and the water content was low in the OS after the treatment. This is consistent with the results shown in Figure 4b. The absolute oil content of the sample was lower than that of the sample treated with SL after removing the moisture. The S/D pyrolysis weight loss yield is lower (36.61%). This indicated that the highest oil removal efficiency was obtained. In fact, the heat loss of OS was mainly divided into three stages. In the first stage, the mass loss was not obvious, and it mainly occurred within 170 °C. This was caused by the loss of water. In the second stage, the quality loss of OS was attributed to the degradation of PHCs. It mainly occurred in the temperature range of 170–670 °C. Figure 4b shows that the original OS has three clear degradation peaks. The maximum weight loss yield (35.50%) occurs at 309 °C and 361 °C. This was attributed to the degradation of alkanes and PAHs. The maximum weight loss yield (23.24%) was attributed to the thermal decomposition of the resins and asphaltenes at 450 °C [61]. Only two obvious degradation peaks are observed in the sludge after thermal washing. The SL-treated sludge has a large degradation peak at 170–400 °C, with a corresponding mass loss of 23.27%. The mass loss is 15.02% in the range of 400–670 °C. The content of asphaltene decreased by 19% compared with the original OS. The decomposition temperature range of the light oils (saturated hydrocarbons and aromatic hydrocarbons) widens. The maximum weight loss yield occurred at 320 °C. This was attributed to the fact that the oil components were fully emulsified in the surfactant. A uniform mixed system formed, and the boiling point of the oil component changed [62]. OS attained the maximum weight loss yield at 343 °C and 440 °C after the DL treatment. The mass losses were 14.73% and 9.57%, respectively. The quality loss of OS treated with S/D was the lowest. The mass change at 170–400 °C was 14.38%, and the mass loss at 400–670 °C was 10.70%. This illustrates that saturated hydrocarbons and aromatic hydrocarbons were effectively removed by adsorption demulsification. In addition, the macromolecular network structure of asphaltenes and resins was effectively destroyed [29]. The third stage was in the temperature range of 640–800 °C. All OS samples showed a small amount of degradation. This part of the mass loss was the degradation of carbonate inorganic substances in the soil [63]. The addition of calcium oxide activated sodium persulfate during the DL and S/D treatments, and the mass losses of the DL and S/D samples were significantly higher than those of the OS and SL treatments.

The gas was collected during the degradation of the OS, and the removal mechanism of PHCs was analyzed. 3D-thermogravimetric infrared of different oily sludge samples is shown in Figure 4c. The OMNIC Spectroscopy software treatment of the absorbance is converted to the transmittance. The two-dimensional infrared spectra are shown in Figure 4d,e. H_2_O formed at temperatures below 170 °C, and this was attributed to the evaporation of water from the OS. Figure 4d shows that the original OS produced H_2_O (3900–3400 cm^−1^, 1960–1400 cm^−1^), CO_2_ (3359 cm^−1^, 2300 cm^−1^, 660 cm^−1^), SO_2_ (1373 cm^−1^, 1327 cm^−1^), CO (2239 cm^−1^, 2197 cm^−1^), NH_3_ (960 cm^−1^, 927 cm^−1^), and the stretching vibration of the C–H bond of alkanes (2930 cm^−1^ and 2855 cm^−1^) after pyrolysis for 20 min (300 °C). The gas was continuously released at temperatures between 280 and 420 °C. The results proved that the original OS contained a large amount of alkanes, and PAHs, and a small amount of sulfur and amines [64]. The SO_2_ peak disappeared after thermal washing. The results showed that the sulfur and amines in the sludge were completely removed. The peak intensity of H_2_O in the S/D sample was lower than that in the SL- and DL-treated samples. This indicates that most alkanes were removed. The peaks of SO_2_, NH_3_, and CO disappear completely after pyrolysis for 30 min, as shown in Figure 4e. The peak value of CO_2_ in the gas products produced in SL attained the highest point at 450 °C. In contrast, the peak value of CO_2_ in the S/D was relatively low. This is attributed to the decomposition of the heavy components in petroleum hydrocarbons. The results showed that large amounts of asphaltenes and resins were removed by adsorption demulsification. A sharp CO_2_ peak appeared at temperatures higher than 600 °C. This is attributed to the pyrolysis of carbonate [61]. This is consistent with the TGA results mentioned above. Therefore, adsorption demulsification was the most effective treatment method, which resulted in the efficient removal of saturated and aromatic hydrocarbons. In addition, resins and asphaltenes were removed by decomposition into small-molecule organic compounds.

### 2.5. Removal Mechanism of PHCs from the Oily Sludge

At present, the main purpose of sludge treatment is to recover and utilize oil components. Therefore, the recovery efficiency of PHCs in OS was analyzed. The separation mechanism of OS was also analyzed. The results are presented in Figure 5. The oil removal efficiency of the SL treatment was 75.59%, but the recovery efficiency of PHCs was only 55.97% (Figure 5a). This was attributed to the oil in the W/O emulsion in the sludge that had not been separated. The oil removal efficiency of the DL treatment was 62.27%. However, the recovery efficiency was 65.89%. The results show that a high recovery efficiency of the OS was attained with demulsification treatment. The oil removal and recovery efficiency were as high as 91.28% and 89.65%, respectively, in the S/D OS. This shows that the multiphase stability system was destroyed through adsorption and demulsification. Both effects, high oil removal and recovery efficiency, could not be achieved using a single treatment. The three-phase separation effect of different samples after standing for 7 days is shown in Figure 5b. An obvious three-phase separation is observed after the S/D treatment. The water layer was clear and transparent. The oil removal efficiency was low, only 62.27%, although the DL treatment had a certain effect of three-phase separation. Thus, the adsorption demulsification treatment resulted not only in a high oil removal efficiency, but also a high recovery efficiency. The SL concentration was approximately 0.55~0.63 g·L^−1^ in the water layer, measured using a UV-visible spectrophotometer. Therefore, the recovered aqueous solution could be recycled, and the waste of resources and secondary pollution were effectively suppressed.

Figure 5c shows the mechanism of separation of the OS during adsorption demulsification. The oil agglomeration (yellow) was covered on the sediment surface (brownish black). First, saturated and aromatic hydrocarbons were adsorbed by the monolayer adsorption of SL on the sediment surface. The micellar particles formed through the crimping action of SL molecules, PHCs were encapsulated in the hydrophobic core by micellar particles, and stubborn and cohesive oil agglomeration was dissociated. However, it was difficult to reflect the effect of the three-phase separation because of the existence of the W/O structure. Numerous SO_4_^−^· were formed from sodium persulfate, and the oxidation reaction between SO_4_^−^· and PAHs took place by hydrogen extraction, addition of unsaturated bonds, and electron transfer pathway [60]. The W/O emulsion was broken, and a good three-phase separation effect was achieved, resulting in significant increases in the oil removal and oil recovery efficiency.

## 3. Materials and Methods

### 3.1. Materials

OS was obtained from the Daqing Field (Daqing, Heilongjiang, China). The water, oil, and solid contents were 13.76%, 36.30%, and 49.94%, respectively. SL was purchased from Sigma-Aldrich (Milwaukee, WI, USA), the sulfonation degree was 0.6 mmol/g, *M*_w_ = ~52,000, Mn = ~7000. Sodium persulfate, sodium hydroxide, and calcium oxide were purchased from Aladdin Reagent Co., Ltd. (Shanghai, China).

### 3.2. Thermal Washing of Oily Sludge

The OS was separated by S/D treatment. OS (2.0 g) was added to a 150 mL beaker. The SL solution was then added to the OS. The pH of the mixture was adjusted using NaOH solution. The OS was treated under different pHs, SL concentrations, temperatures, solid–liquid ratios, revolving speed, and SL treatment times. Demulsification was performed immediately after the SL treatment. Sodium persulfate was activated using calcium oxide (molar ratio 1.5:1.0). The effects of sodium persulfate dosage, demulsification temperature, and demulsification time on the oil removal efficiency were analyzed. The three-phase separation of oil, water, and mud sand was realized 2.0 h after the reaction. Three sets of parallel experiments were performed under the same conditions. The concentration of SL in the aqueous phase of OS was calculated after static delamination was measured by ultraviolet and visible spectrophotometer (UV/VIS, SPECORD-PLUS-50, Analytik Jena, Germany). The oil content, water content, and soil content were analyzed by an automatic washing and drying system (prepASH 340, Precisa, Switzerland), and the temperature rising range was 25–600 °C. The quality loss of water was at 25–105 °C, and oil quality at 105–600 °C; the remaining material was soil and inorganic salt. The oil removal efficiency and oil recovery efficiency were calculated [11] as follows:Q_R1_ (%) = (m_1_ − m_2_)/m_1_ × 100%(1)
where Q_R1_ was the removal efficiency of oily sludge (%), m_1_ was the oil content of the original oily sludge (g), and m_2_ was the oil content of the sludge after treatment (g).
Q_R2_ (%) = m_3_/m_1_ × 100%(2)
where Q_R2_ was the recovery efficiency of oily sludge (%), m_1_ was the oil content of the original oily sludge (g), and m_3_ was the extracted oil content after treatment (g).

The efficiency of oil removal using traditional SL treatment and DL treatment were compared. Traditional SL treatment was performed according to the method reported by Ma et al. [24]. The specific method and process of DL treatment were described by Zhao et al. [65]. The oil removal efficiency, oil recovery efficiency, and concentration of SL in the aqueous phase of OS were calculated after static delamination [11].

### 3.3. Physicochemical Properties of Oily Sludge

The apparent morphology of OS after different treatments was observed through SEM (SU8220, Hitachi, Tokyo, Japan). The samples were observed under 10 kV voltage after spraying gold in a vacuum for 90 s [66]. The hydrophobicity of the soil surface in the OS was analyzed using a drop shape analyzer (DSA100, KRUSS, Hamburg, Germany) [67]. The average value was obtained from multiple measurements.

The viscosities of the samples were analyzed using a combined rheometer workstation (HAAKE MARS4, Waltham, MA, USA). The sample was poured onto the test platform of the modular rheometer workstation at 25 °C. The interval between the rotor and test platform was 1 mm. The solid content of the sample was 30%. Viscosity is the ratio of the shear stress of the sample [58], and was calculated as follows:η = τ/γ(3)
where η is the viscosity (Pa·s), τ is the shear stress (Pa), and γ is the shear rate (1/s).

The main functional groups of PHCs on the soil surface of OS were analyzed using FTIR (TENSEOR27, Bruker, Germany). Potassium bromide and solid sludge particles were fully mixed and pressed [68]. The spectral range was 4000–400 cm^−1^ and the resolution was 2 cm^−1^.

The thermal degradation products of different OS samples were analyzed through TGA (TGA55, TA Instruments, USA)/FTIR (Thermo Fisher Scientific, Waltham, MA, USA). The OS (10 mg) was heated at 30–800 °C for 15 °C·min^−1^. The nitrogen flow rate was 60 mL·min^−1^ [69]; the temperature of the hot wire and infrared battery was 250 °C; the infrared spectrum range was 4000–500 cm^−1^; the resolution was 4 cm^−1^; and four scans were carried out.

## 4. Conclusions

S/D treatment, efficiently combining adsorption and demulsification, was found to be a green and efficient separation method for OS. The oil removal and recovery of OS were as high as 91.28 and 89.65%, respectively. Saturated and aromatic hydrocarbons in OS were adsorbed by SL. The solubilization effect of PHCs was attained, and the oxidative demulsification effect of sodium persulfate was effectively demonstrated. PAHs were oxidized and degraded into small-molecule organic compounds. The macromolecular network structure between asphaltenes and colloids was destroyed. A good three-phase separation effect was observed. This indicated that the adsorption demulsification treatment had a strong capability to remove PHCs and attained a better viscosity reduction effect. This provides theoretical support for the efficient separation of OS.

## Figures and Tables

**Figure 1 ijms-23-07504-f001:**
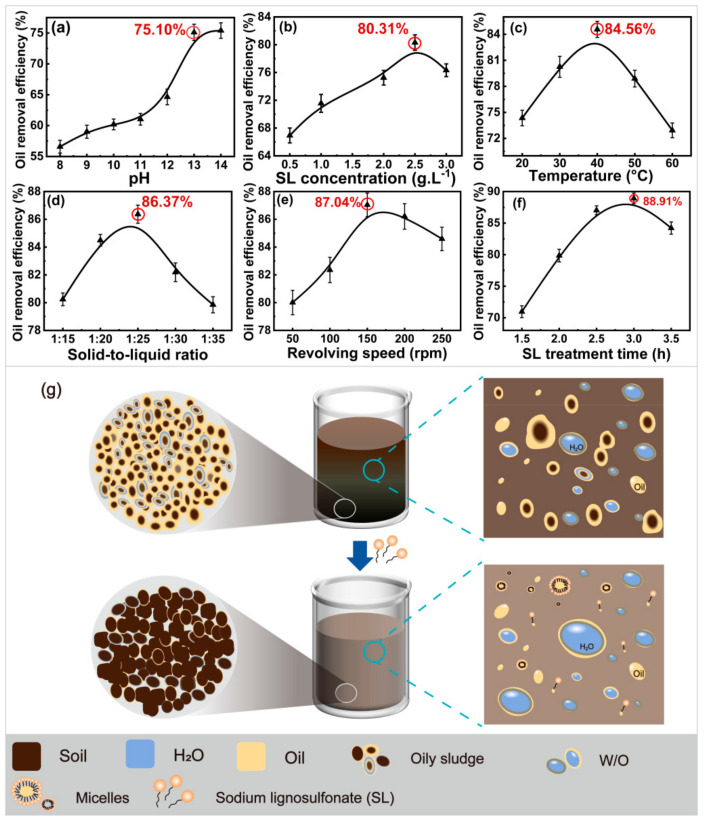
Effects of adsorption on the oil removal efficiency in S/D treatment. ((**a**), pH; (**b**), SL concentration; (**c**), temperature; (**d**), solid–liquid ratio; (**e**), revolving speed; (**f**), time; (**g**), adsorption mechanism of sodium lignosulfonate).

**Figure 2 ijms-23-07504-f002:**
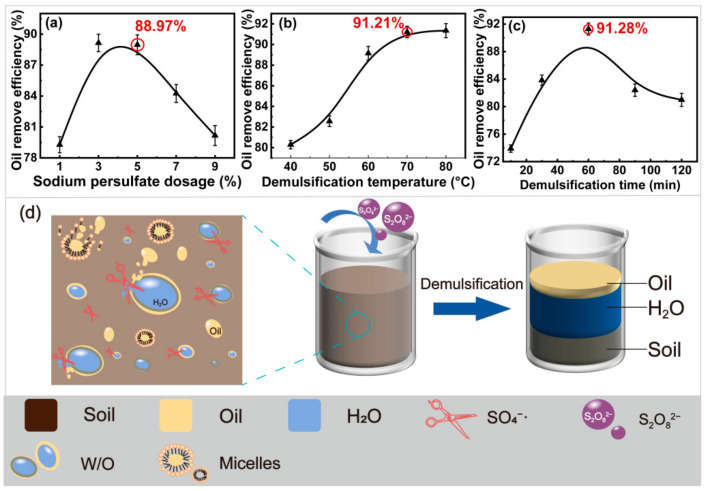
Effects of demulsification on oil removal efficiency in S/D treatment. ((**a**), sodium persulfate dosage; (**b**), demulsification temperature; (**c**), demulsification time; (**d**), demulsification mechanism of sodium persulfate).

**Figure 3 ijms-23-07504-f003:**
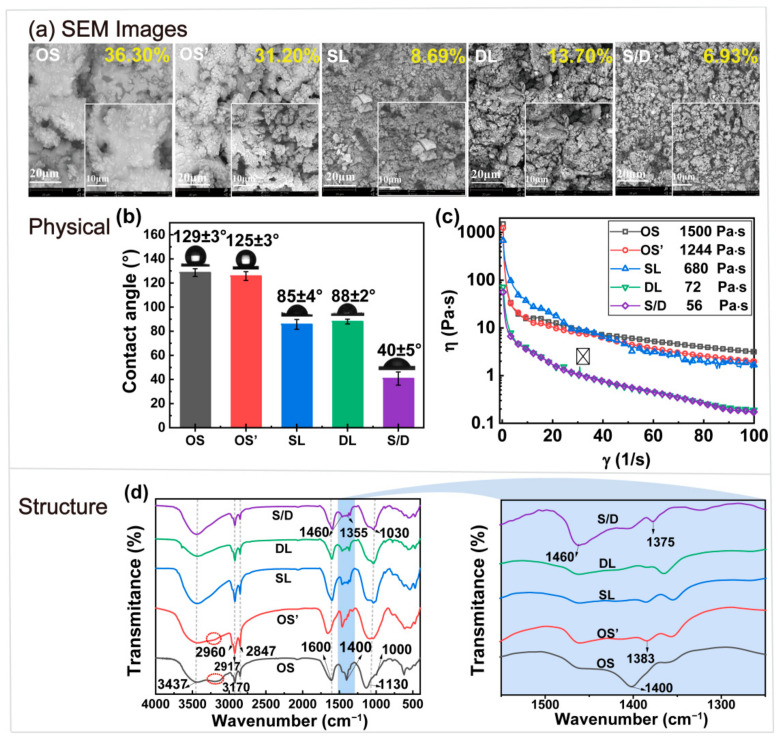
Effect of different treatment methods on physical and chemical characteristics of oily sludge (OS) ((**a**), SEM of OS; (**b**), Contact angle of OS; (**c**), OS viscosity; (**d**), FITR of OS).

**Figure 4 ijms-23-07504-f004:**
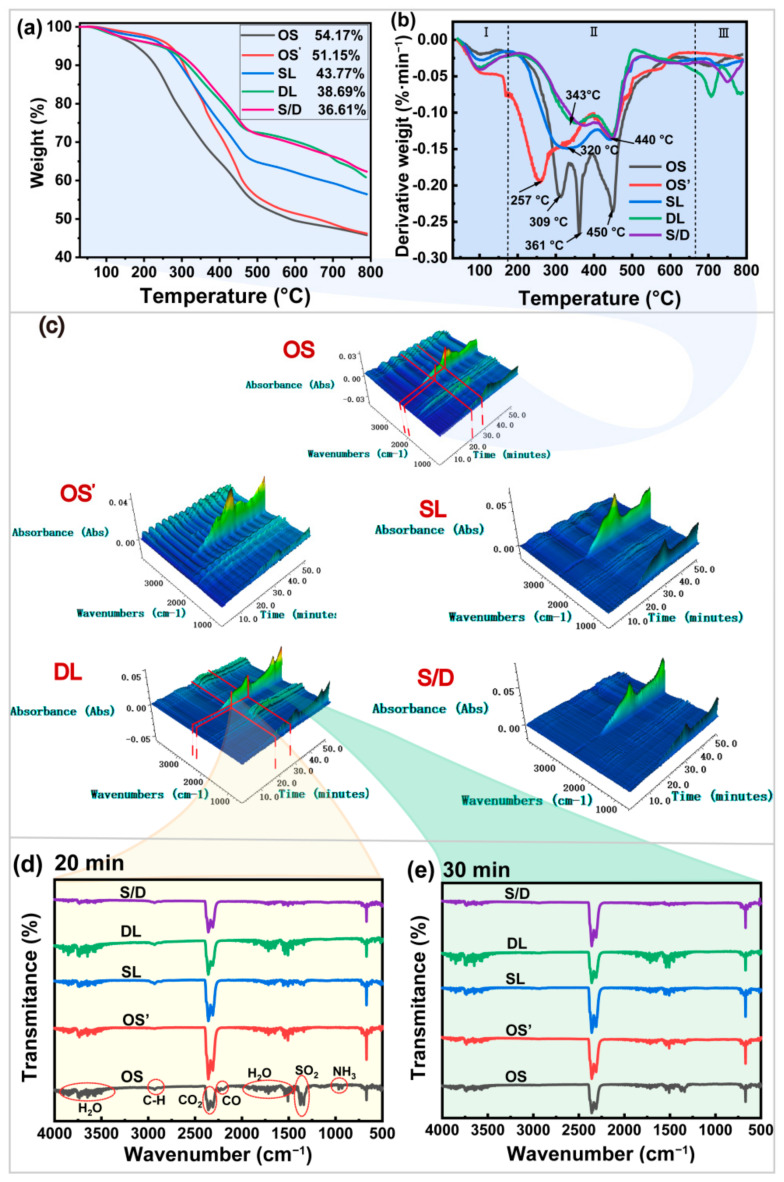
TG-FITR of OS after different treatments. ((**a**), TG curves; (**b**), DTG curves; (**c**), 3D stacked plot diagram of evolved gases. (**d**), IR spectra of different OS after 20 min of pyrolysis (300 °C); (**e**), FTIR spectra of different OS after 30 min of pyrolysis (450 °C)).

**Figure 5 ijms-23-07504-f005:**
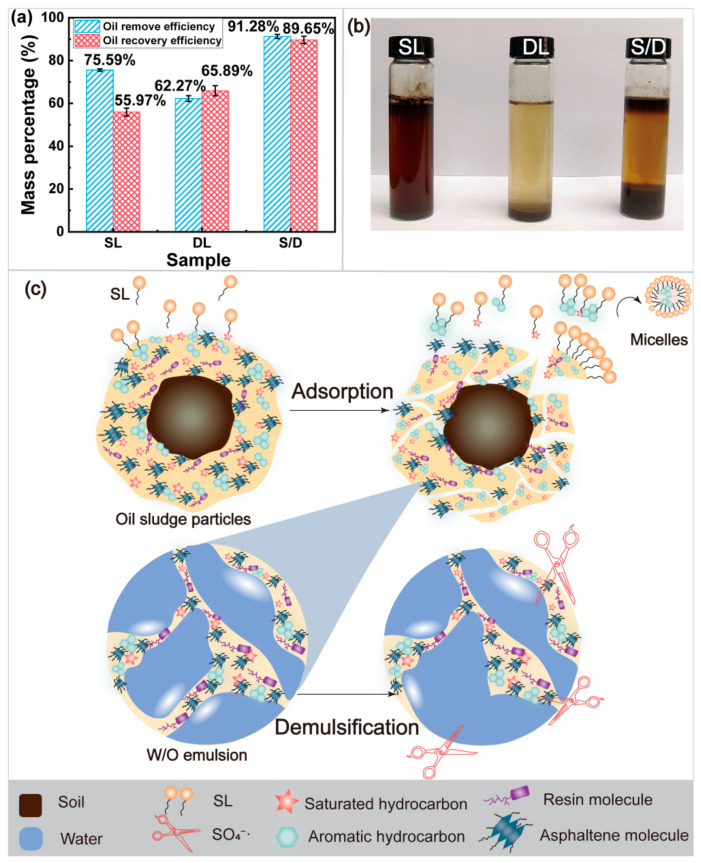
Recovery and removal mechanisms of petroleum hydrocarbons from the OS ((**a**), removal and recovery of petroleum hydrocarbons from the OS; (**b**), digital photographs were processed by different methods after standing for 7 d; (**c**), removal mechanism of petroleum hydrocarbons from the OS).

## Data Availability

The data presented in this study are available in the manuscript’s figure.

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
