# Peer review of "Efficient Separation and Recovery of Petroleum Hydrocarbon from Oily Sludge by a Combination of Adsorption and Demulsification"

_ijms, 2022, doi:10.3390/ijms23147504_

Round 1
Reviewer 1 Report
For the separation and recovery of petroleum hydrocarbons from oily sludge, a fast and low-cost method is required. The authors investigated and discussed a simple and effective method via a combination of adsorption and demulsification. Sodium lignosulfonate (SL) concentration, pH, temperature, solid–liquid ratio, revolving speed, and time on SL adsorption solubilization and sodium persulfate dosage, demulsification temperature, and demulsification time on sodium persulfate oxidative demulsification were investigated. The manuscript is well written, and I would advise publication after minor revision.
(1) The authors should do a cost analysis of the strategy. Since oily sludge is of low economic value, the authors need to convince readers that the process to recover hydrocarbon has an economic interest.
(2) Figure 1
The authors used “oil removal yield” which is a strange word and confusing. The authors need to revise the graph and discussion on “oil removal yield”.
Figure 2 is vague. The picture quality needs to be improved.
(3) Page 9, line 378-382
Please correct typos in the discussion on Figure 3d.
(4) Figure 4
Please pay attention to the Y-axis for IR signals. In Figure 4c, the absolute intensity of IR signal were used; in figure 4d, seem the normalized value were recorded. The authors need to clarify in figures.
(5) The obvious weakness of the manuscript is that only a single factor effect analysis was performed. Parameters, such as surfactant amount, reaction temperature, and treatment time are usually correlated. I hope in the future the authors would consider performing the analysis/optimization using statistical strategies; the factorial design, linear regression, and response surface are useful tools for these kinds of work.
Author Response
Dear Reviewer,
Thank you for your letter and for the comments concerning our manuscript entitled “Efficient separation and recovery of petroleum hydrocarbon from oily sludge by a combination of adsorption and demulsification”. We have studied your comments carefully and have made corrections which we hope could meet your requirements. All changes were marked up using the “Track Changes” function.
Questions you put forward are explained as follows:
(1) The authors should do a cost analysis of the strategy. Since oily sludge is of low economic value, the authors need to convince readers that the process to recover hydrocarbon has an economic interest.
In fact, the main advantage of oily sludge separation is to solve soil pollution. The treated soil can be directly used in industrial and agricultural production. In addition, the separated oil can be recycled. As a by-product of biomass refining, sodium lignosulfonate is used to heat wash oily sludge. It does not enhance the added value of lignocellulosic biomass, but has high environmental value and economic feasibility.
(2) Figure 1, the authors used “oil removal yield” which is a strange word and confusing. The authors need to revise the graph and discussion on “oil removal yield”. Figure 2 is vague. The picture quality needs to be improved.
In fact, “oil removal yield”was replaced by “oil removal efficiency” in Figure 1. And the graph and discussion were reviewed and modified in the whole manuscript.
The picture quality of Figure 2 has been improved.
(3) Page 9, line 378-382, Please correct typos in the discussion on Figure 3d.
The correct typos in the discussion on Figure 3d was modified.
Fig. 4d shows that the original OS produced H2O (3900–3400 cm-1, 1960–1400 cm-1), CO2 (3359 cm-1, 2300 cm-1, 660 cm-1), SO2 (1373 cm-1, 1327 cm-1), CO (2239 cm-1, 2197 cm-1), NH3 (960 cm-1, 927 cm-1), and the stretching vibration of the C–H bond of alkanes (2930 cm-1 and 2855 cm-1) after pyrolysis for 20 min (300 °C).
(4) Figure 4, Please pay attention to the Y-axis for IR signals. In Figure 4c, the absolute intensity of IR signal were used; in figure 4d, seem the normalized value were recorded. The authors need to clarify in figures.
The Y-axis for IR signals is absorbance in Figure 4c. We use the OMNIC Spectroscopy software to process the raw data and convert the Y-axis to the transmittance, so the Y-axis is the transmittance in Figure 4d. This kind of data expression way is a common data analysis, is convenient for the reader to read. We revised and explained the manuscript as follows:
3D-thermogravimetric infrared of different oily sludge samples is shown in Fig. 4c. The OMNIC Spectroscopy software treatment of the absorbance is converted to the trans-mittance. The two-dimensional infrared spectra are shown in Fig. 4d and 4e.
(5) The obvious weakness of the manuscript is that only a single factor effect analysis was performed. Parameters, such as surfactant amount, reaction temperature, and treatment time are usually correlated. I hope in the future the authors would consider performing the analysis/optimization using statistical strategies; the factorial design, linear regression, and response surface are useful tools for these kinds of work.
We understand that the analysis/optimization using statistical strategies; the factorial design, linear regression, and response surface may better reveal the interaction of factors. However, in the present study, we mainly focused on a new method of adsorption-demulsification. The separation effect of oily sludge was analyzed through sodium lignosulfonate (SL)-assisted sodium persulfate (S/D) treatment. And we think that the oil removal yield as high as 91.28% in the study may not be optimal, but should be sufficient to draw a conclusion that S/D treatment, efficiently combining adsorption and demulsification, was found to be a green and efficient separation method for oily sludge. The analysis/optimization using statistical strategies; the factorial design, linear regression, and response surface are under investigation in our laboratory. Unfortunately, results are unavailable at this point.
Reviewer 2 Report
This manuscript investigates the recovery of petroleum hydrocarbon from oily sludge by a combination of adsorption and demulsification and analyzes the effects of sodium lignosulfonate concentration, pH, temperature, solid–liquid ratio, revolving speed, and time on sodium lignosulfonate adsorption solubilization. Overall, this manuscript can be published in IJMS.
Author Response
Dear Reviewer,
Thank you for your letter and for the comments concerning our manuscript entitled “Efficient separation and recovery of petroleum hydrocarbon from oily sludge by a combination of adsorption and demulsification”. We have studied your comments carefully and have made corrections which we hope could meet your requirements. All changes were marked up using the “Track Changes” function.
This manuscript investigates the recovery of petroleum hydrocarbon from oily sludge by a combination of adsorption and demulsification and analyzes the effects of sodium lignosulfonate concentration, pH, temperature, solid–liquid ratio, revolving speed, and time on sodium lignosulfonate adsorption solubilization. Overall, this manuscript can be published in IJMS.
Thank you very much. It is a great honor to be recognized by you for this work. We have double-checked our manuscripts to make sure that they are correct. Thank you again for your recognition of our work.
Reviewer 3 Report
Recommendation: Minor revisions needed as noted
Comments:
The manuscript entitled “Efficient separation and recovery of petroleum hydrocarbon from oily sludge by a combination of adsorption and demulsification” should be an interesting paper to the readers. This work showed the separation of oil from oily sludge which helps to clean environmental pollution and to collect some oil energy. In this manuscript, Yao et al. have studied the effect of sodium lignosulfonate assisted sodium persulfate under different conditions such as pH, temperature and etc.
The results merit to publish in molecular sciences journal. Overall, the paper provides enough work to show how these chemicals work to remove oil. I would recommend its publication after the following minor questions are properly addressed in the revised manuscript.
1. It would be better to study this work with at least two different oily sludges to make sure these chemicals work with different oily sludges.
Other comments:
1. In line 13, S/D is given as short form of chemical. Is it sort form of sodium persulfate or SL assisted sodium persulfate. Why D instead of P?
2. In lines 21, 22 and in few other lines there is period after SO4-. What is the significance of the period there? It should be corrected.
3. The oil recovery is given in percentage. How do authors know how much oil was originally there in the oil sludge?
4. Is the OS viscosity very high? Just curious how trace amount of oil in the brine gives such a high viscosity of about 1500 Pa.s?
5. In line 74, authors have written that the synergistic effects of SL and sodium persulfate were also analyzed. How do authors know it is only combined effect or synergistic effect.
6. In line 80, Sodium persulfate is represented as DL. What is the difference between DL and S/D for the same sodium persulfate. It gives confusion to the authors.
7. In the Figure 1, the oil removal yield vs at different conditions are plotted. Why the oil removal yield started at around 55% yield? Why not lower than 55% or why it has not started from 0?
8. The study was done with different percent of sodium persulfate dosage from 1-9% and the optimum does is 5%. It increases the oil amount from about 79% to 88%. Is 5% sodium persulfate dosage practical meaning economical to use?
9. In all the references there is comma at the end of the reference. There should be period instead of comma. Make sure this is in correct format.
10. In most of the references the first letter of the title of the paper is capital but if new references such as in ref. 68 first letter of each work is capital. This should be consistent in all the references.
Author Response
Dear Reviewer,
Thank you for your letter and for the comments concerning our manuscript entitled “Efficient separation and recovery of petroleum hydrocarbon from oily sludge by a combination of adsorption and demulsification”. We have studied your comments carefully and have made corrections which we hope could meet your requirements. All changes were marked up using the “Track Changes” function.
Questions you put forward are explained as follows:
1. It would be better to study this work with at least two different oily sludges to make sure these chemicals work with different oily sludges.
We understand that studied different oily sludges to make sure these chemicals work with different oily sludges may better reveal applicability of the method.
In this study, we mainly focused on a new method of adsorption-demulsification. The separation effect of oily sludge was analyzed through sodium lignosulfonate (SL)-assisted sodium persulfate (S/D) treatment. And we think that the oil removal yield as high as 91.28% in the study may not be optimal, but should be sufficient to draw a conclusion that S/D treatment, efficiently combining adsorption and demulsification, was found to be a green and efficient separation method for oily sludge. In fact, oily sludge is mainly divided into ground sludge, oil-based drilling cuttings, refining sludge and tank bottom sludge. The treatment of oily sludge varies with species. Treatment conditions of oily sludge vary with species. The results show that S/D treatment has a good separation effect on the ground sludge. The separation effect of S/D treatment on different oily sludge is studied in our follow-up study.
2. In line 13, S/D is given as short form of chemical. Is it sort form of sodium persulfate or SL assisted sodium persulfate. Why D instead of P?
S/D is abbreviation of the name of oily sludge sample treated with SL-assisted sodium persulfate. The explanation of S/D is added in the "Abbreviations" section at the bottom of the manuscript (Lines 528-532), which is as follows:
Abbreviations
OS: Oily sludge
SL: Sodium lignosulfonate
DL: Sodium persulfate treatment
S/D: Sodium lignosulfonate-assisted sodium persulfate treatment
3. In lines 21, 22 and in few other lines there is period after SO4-. What is the significance of the period there? It should be corrected.
Incorrect description has been modified. Lines 20-22 have been revised as follows:
Sulfate radical (SO4-·) with a high oxidation potential, was formed from sodium persulfate.
4. The oil recovery is given in percentage. How do authors know how much oil was originally there in the oil sludge?
The details were added in the revision (Lines 473-475).
The oil content, water content and soil content were analyzed by automatic ashing and drying system (prepASH 340, precisa, Switzerland). The untreated oily sludge was put into the automatic ashing and drying system, and the temperature rising range was 25-600 °C. The quality loss of water at 25-105 ℃, and oil quality at 105-600 ℃, the remaining material is soil and inorganic salt.
5. Is the OS viscosity very high? Just curious how trace amount of oil in the brine gives such a high viscosity of about 1500 Pa.s?
The oily sludge (OS) viscosity is very high. There are a lot of asphaltenes (8-10%) and resins (7-22.4%) in oily sludge. Due to the volatilization of water molecules, light components (saturated hydrocarbons, aromatic hydrocarbons) volatilization. As a result, the total oil content of oily sludge increased to 36.3% (Lines 441 in manuscript). This improves the viscosity and stability of oily sludge.
6. In line 74, authors have written that the synergistic effects of SL and sodium persulfate were also analyzed. How do authors know it is only combined effect or synergistic effect.
S/D processing is much more efficient than SL and DL in the analysis of part 2.3 of the manuscript. The oil content, contact angle and initial viscosity are significantly reduced by S/D treatment. Such as The contact angle of SL and DL were 85° and 88°, respectively. This means that the separation effects of the traditional SL treatment and DL treatment were very similar. The contact angle of the S/D-treated sludge significantly decreased (40°). The sludge exhibited a strongly hydrophilic surface in Fig.3b. This proves that the synergistic effects of SL and sodium persulfate. The mechanism of synergy is analyzed in part 2.4 of the manuscript.
7. In line 80, Sodium persulfate is represented as DL. What is the difference between DL and S/D for the same sodium persulfate. It gives confusion to the authors.
DL is sodium persulfate treatment without sodium lignosulfonate. S/D is sodium lignosulfonate-assisted sodium persulfate treatment. The explanation of OS, SL, DL and S/D are added in the "Abbreviations" section at the bottom of the manuscript (Lines 528-532), which is as follows:
Abbreviations
OS: Oily sludge
SL: Sodium lignosulfonate
DL: Sodium persulfate treatment
S/D: Sodium lignosulfonate-assisted sodium persulfate treatment
8. In the Figure 1, the oil removal yield vs at different conditions are plotted. Why the oil removal yield started at around 55% yield? Why not lower than 55% or why it has not started from 0?
Sodium lignosulfonate is suitable for alkaline environment, so its treatment is studied under alkaline condition. The study of acidic conditions is not meaningful, and weak base conditions (pH = 8), oil removal rate of 55%. Therefore, this shows that the method is effective.
9. The study was done with different percent of sodium persulfate dosage from 1-9% and the optimum does is 5%. It increases the oil amount from about 79% to 88%. Is 5% sodium persulfate dosage practical meaning economical to use?
The main aim of studying the dosage of sodium persulfate is to prevent the quenching reaction of SO4-·. If sodium persulfate is added in excess, a large amount of SO4-· and OH· free radicals quickly will form in the excess sodium persulfate solution. It was quenched to failure as the high local concentration of SO4-· promoted mutual free radicals. The eluted heavy oil was reabsorbed onto the solid surface.
10. In all the references there is comma at the end of the reference. There should be period instead of comma. Make sure this is in correct format.
The format of the references has been modified as suggested.
11. In most of the references the first letter of the title of the paper is capital but if new references such as in ref. 68 first letter of each work is capital. This should be consistent in all the references.
The following references were corrected. These were [17], [18], [19], [24], [25], [36], [53], [58], [60], [62], [67], [68].